# Predictive Value of Modified Glasgow Prognostic Score and Persistent Inflammation among Patients with Non-Small Cell Lung Cancer Treated with Durvalumab Consolidation after Chemoradiotherapy: A Multicenter Retrospective Study

**DOI:** 10.3390/cancers15174358

**Published:** 2023-09-01

**Authors:** Keiko Tanimura, Takayuki Takeda, Akihiro Yoshimura, Ryoichi Honda, Shiho Goda, Shinsuke Shiotsu, Mototaka Fukui, Yusuke Chihara, Kiyoaki Uryu, Shota Takei, Yuki Katayama, Makoto Hibino, Tadaaki Yamada, Koichi Takayama

**Affiliations:** 1Department of Respiratory Medicine, Japanese Red Cross Kyoto Daini Hospital, Kyoto 602-8026, Japan; keiko-t@koto.kpu-m.ac.jp (K.T.); aki-y@koto.kpu-m.ac.jp (A.Y.); 2Department of Respiratory Medicine, Asahi General Hospital, Asahi 289-2511, Japan; honda_r1@icloud.com; 3Department of Respiratory Medicine, Japanese Red Cross Kyoto Daiichi Hospital, Kyoto 605-0981, Japan; oomtsh25kgs@yahoo.co.jp (S.G.); sshiotsu@gmail.com (S.S.); 4Department of Respiratory Medicine, Uji-Tokushukai Medical Center, Uji 611-0041, Japan; fukuimo@koto.kpu-m.ac.jp (M.F.); c1981311@koto.kpu-m.ac.jp (Y.C.); 5Department of Respiratory Medicine, Yao Tokushukai General Hospital, Yao 581-0011, Japan; kiyoaki.uryuu@tokushukai.jp; 6Department of Pulmonary Medicine, Graduate School of Medical Science, Kyoto Prefectural University of Medicine, Kyoto 602-8566, Japan; tki-sho@koto.kpu-m.ac.jp (S.T.); ktym2487@koto.kpu-m.ac.jp (Y.K.); tayamada@koto.kpu-m.ac.jp (T.Y.); takayama@koto.kpu-m.ac.jp (K.T.); 7Department of Respiratory Medicine, Shonan Fujisawa Tokushukai Hospital, Fujisawa 251-0041, Japan; makotohibino560328@yahoo.co.jp

**Keywords:** non-small cell lung cancer, durvalumab consolidation, chemoradiotherapy, inflammation, modified Glasgow prognostic score, C-reactive protein

## Abstract

**Simple Summary:**

Durvalumab consolidation after chemoradiotherapy (CRT) is a standard treatment for locally advanced non-small cell lung cancer, which sometimes encounters early recurrence. This retrospective study aimed to identify the predictors of durvalumab consolidation after CRT. A prognostic risk classification was created combining modified Glasgow Prognostic Score (mGPS) before CRT and C-reactive protein (CRP) level after CRT. When patients with pre-CRT mGPS of 0 or mGPS of 1 with post-CRT CRP ≤1 mg/dL were classified as the “low-risk” group, and patients with pre-CRT mGPS of 2 or mGPS of 1 with post-CRT CRP >1 mg/dL were classified as the “high-risk” group, the high-risk group had a significantly shorter median progression-free survival (PFS, hazard ratio [HR]: 2.47, *p* < 0.001) and overall survival (OS, HR: 3.62, *p* < 0.001) compared with those in the low-risk group. The prognostic risk classification helps to predict the PFS and OS of durvalumab consolidation after CRT.

**Abstract:**

Background: Durvalumab consolidation after chemoradiotherapy (CRT) is a standard treatment for locally advanced non-small cell lung cancer (NSCLC). However, studies on immunological and nutritional markers to predict progression-free survival (PFS) and overall survival (OS) are inadequate. Systemic inflammation causes cancer cachexia and negatively affects immunotherapy efficacy, which also reflects survival outcomes. Patients and Methods: We retrospectively investigated 126 patients from seven institutes in Japan. Results: The modified Glasgow Prognostic Score (mGPS) values, before and after CRT, were the essential predictors among the evaluated indices. A systemic inflammation-based prognostic risk classification was created by combining mGPS values before CRT, and C-reactive protein (CRP) levels after CRT, to distinguish tumor-derived inflammation from CRT-induced inflammation. Patients were classified into high-risk (*n* = 31) and low-risk (*n* = 95) groups, and the high-risk group had a significantly shorter median PFS of 7.2 months and an OS of 19.6 months compared with the low-risk group. The hazard ratios for PFS and OS were 2.47 (95% confidence interval [CI]: 1.46–4.19, *p* < 0.001) and 3.62 (95% CI: 1.79–7.33, *p* < 0.001), respectively. This association was also observed in the subgroup with programmed cell death ligand 1 expression of ≥50%, but not in the <50% subgroup. Furthermore, durvalumab discontinuation was observed more frequently in the high-risk group than in the low-risk group. Conclusion: Combining pre-CRT mGPS values with post-CRT CRP levels in patients with locally advanced NSCLC helps to predict the PFS and OS of durvalumab consolidation after CRT.

## 1. Introduction

Consolidation therapy with durvalumab, an anti-programmed cell death ligand 1 (PD-L1) antibody, has become the standard treatment after chemoradiotherapy (CRT) in patients with unresectable locally advanced non-small cell lung cancer (NSCLC) [1]. However, some patients experience early recurrence despite receiving durvalumab consolidation after CRT. Therefore, a predictive marker for early recurrence after durvalumab consolidation is essential for detecting recurrence without delay, enabling subsequent treatment.

PD-L1 expression on tumor cells and the tumor mutational burden are crucial factors determining the therapeutic efficacy of immune checkpoint inhibitors (ICIs) for advanced malignancies [2,3]. They are essential among patients who undergo durvalumab consolidation after CRT in locally advanced stages [4].

In addition, the tumor microenvironment, with its substantial cytokine functions, is a factor that defines tumor cell behavior in the cancer-immunity cycle. Pro-inflammatory cytokines such as interleukin (IL)-6 and IL-1β promote tumor proliferation and suppress immune responses to tumor cells, contributing to tumor progression [5,6,7]. These inflammatory cytokines are also involved in cancer cachexia development—defined as “a multifactorial syndrome defined by an ongoing loss of skeletal muscle mass (with or without loss of fat mass) that cannot be fully reversed by conventional nutritional support and leads to progressive functional impairment [8]”. Cancer cachexia is reported in over half of the patients with advanced cancer [8]. In addition, pre-cachexia or cachexia phases are even observed in patients with early or locally advanced stages. Studies have reported that systemic inflammation precedes typical cachexia criteria, such as muscle wasting and body weight loss [9,10]. Cancer cachexia is associated with poor prognosis in immunotherapy through the desensitization of programmed cell death-1 (PD-1)/PD-L1 inhibition [11,12]. Therefore, systemic inflammation, which is closely associated with cachexia, could have a prognostic value in durvalumab consolidation after CRT.

Overall survival (OS) in several malignancies can be predicted using immunological and nutritional markers. Potential markers for predicting the prognosis of patients with advanced NSCLC, which are easily calculated from clinical laboratory data and physical measurement, include the neutrophil-to-lymphocyte ratio (NLR) [13,14], platelet-to-lymphocyte ratio (PLR) [15,16], C-reactive protein-to-albumin ratio (CAR) [17,18], advanced lung cancer inflammation index (ALI) [19], systemic immune inflammation index (SII) [20,21], lung immune prognostic index (LIPI) [22], and modified Glasgow prognostic score (mGPS) [23,24]. NLR [25] and CAR [26] predict progression-free survival (PFS) in patients with locally advanced NSCLC who undergo durvalumab consolidation after CRT. In contrast, mGPS can predict the prognosis and evaluate the progression of cachexia based on CRP and albumin levels, which can classify cachexia stages [27]. The influence of CRT-induced inflammation, especially radiation pneumonitis, should be considered when using mGPS to determine the prognosis of patients with advanced NSCLC who undergo durvalumab consolidation after CRT. This application is challenging because it is difficult to differentiate CRT-induced inflammation from tumor inflammation, which is the original significance in calculating the mGPS [23,24]. Therefore, cautious evaluation is required to establish predictive markers for durvalumab consolidation after CRT.

Thus, this multicenter retrospective study aimed to identify the predictors of durvalumab consolidation after CRT, especially focusing on systemic inflammation and cachexia, to detect recurrence without delay.

## 2. Materials and Methods

### 2.1. Patients

We conducted a multicenter retrospective study in Japan to investigate markers for predicting the prognosis of patients with NSCLC who underwent durvalumab consolidation after CRT. We analyzed the medical records of patients with NSCLC who underwent durvalumab consolidation after concurrent CRT between 1 July 2018, and 31 March 2021, at seven institutes in Japan. Inclusion criteria were as follows: (a) aged ≥20 years at the administration of durvalumab; (b) histologically diagnosed NSCLC; (b) durvalumab consolidation after concurrent CRT; (d) evaluable lesion(s) based on the Response Evaluation Criteria in Solid Tumors (RECIST) guidelines (version 1.1). Exclusion criteria were as follows: (a) had been treated with ICIs; (b) judged ineligible to participate in this study by the investigator. The deadline for collecting survival analysis data was 30 September 2021. The study protocol was conducted in accordance with the Declaration of Helsinki and approved by the Ethics Committees of the Japanese Red Cross Kyoto Daini Hospital (8 July 2021; S2021-12) and each participating hospital. The requirement for informed consent was waived because of the retrospective nature of the study. However, patients were allowed to withdraw their data, and the relevant information concerning the study was available on each hospital’s website.

### 2.2. Immunological and Nutritional Markers

Immunological and nutritional markers were calculated at 2 timepoints: at baseline, before CRT, and before durvalumab consolidation, after CRT. The cutoff values were used based on previous reports [13,16,17,19,21,22,24]. CRP values were also investigated after CRT in order to evaluate the persistent inflammation after CRT.

mGPS score: 0 (CRP ≤1 mg/dL and albumin ≥3.5 g/dL), 1 (CRP >1 mg/dL or albumin <3.5 g/dL), or score 2 (CRP >1 mg/dL and serum albumin <3.5 g/dL]); CAR = CRP (mg/dL)/serum albumin (g/dL): grouped based on CAR <0.32 or ≥0.32; NLR = neutrophil count (/µL)/lymphocyte count (/µL): grouped using NLR <5 or ≥5; PLR = platelet count (/µL)/lymphocyte count (/µL): grouped using PLR <180 or ≥180; ALI = Body mass index (kg/m^2^) × serum albumin (g/dL)/NLR: grouped based on ALI ≥18 or <18; SII = platelet count (10^3^/µL) × neutrophil count (/µL)/lymphocyte count (/µL): grouped using SII <750 or ≥750; LIPI: good (derived NLR [dNLR] ≤3 and lactate dehydrogenase [LDH] ≤ upper limit of normal [ULN]), intermediate (dNLR >3 or LDH > ULN), poor (dNLR >3 and LDH > ULN).

### 2.3. Response Evaluation and Outcome Assessment

OS was the interval between the first day of durvalumab consolidation and death from any cause. PFS was the interval from the first day of durvalumab consolidation to disease progression or death, whichever occurred first. Objective response and disease control rates were defined as “the percentage of patients in the study or treatment group who achieved complete response (CR) or partial response (PR) to the treatment” and “the percentage of patients in the study or treatment group who have achieved CR, PR and stable disease (SD)”, respectively [28]. When no imaging/measurement was done after CRT, the patient was not evaluable (NE) [28]. The response was evaluated according to the best overall treatment response based on the RECIST guidelines (version 1.1). Adverse events were assessed in accordance with the common terminology criteria for adverse events (version 5.0).

### 2.4. Statistical Analysis

OS and PFS curves are illustrated using the Kaplan–Meier method. The log-rank test was used to evaluate PFS and OS. Categorical variables were compared using Fisher’s exact test. Cox proportional hazards models were used for univariate or multivariate analyses of PFS and OS. The concordance index (C-index) was used to evaluate the predictive value of markers. Patients with missing data were excluded from the analysis. For all analyses, a *p*-value of <0.05 indicated significance.

### 2.5. Software Tools

Statistical analyses were performed using GraphPad Prism8 (GraphPad Software, San Diego, CA, USA) and EZR statistical software version 1.55 (Saitama Medical Center, Jichi Medical University, Saitama, Japan), which is a graphical user interface for R version 1.61 (R Foundation for Statistical Computing, Vienna, Austria). EZR statistical software is a modified version of the R commander designed to add statistical functions frequently used in biostatistics.

## 3. Results

### 3.1. Characteristics of Patients

In total, 133 patients were enrolled in this study and 7 patients were excluded due to data unavailability (Appendix A). The median duration of follow-up for surviving participants was 16.3 months (95% confidence interval [CI]: 14.7–20.8 months). The baseline characteristics of the 126 patients at the time of durvalumab initiation are summarized in Table 1. The median age was 71 (interquartile range [IQR]: 64.3, 76.0) years, and 98 patients (77.8%) were males. One hundred and ten (87.3%) patients had a smoking history. The numbers of patients with Eastern Cooperative Oncology Group performance statuses (ECOG-PS) of 0–1 and 2 were 118 (93.7%) and 8 (6.3%), respectively. Sixty-five (51.6%) patients had squamous cell carcinoma. The oncogenic driver gene was detected in 8 (6.4%) patients, and ≥50% PD-L1 expression was detected on tumor cells in 41 (32.5%). Fifty-nine (46.8%) patients were in stage IIIA or earlier. The best overall response to CRT was PR in 93 patients (73.8%), SD in 32 (25.4%), and NE in 1 (0.8%). The median interval between CRT and durvalumab consolidation was 15.5 days (IQR, 13–29.75).

### 3.2. Identification of the Most Significant Immunological and Nutritional Marker among Candidate Markers

The immunological and nutritional markers: CAR, NLR, PLR, ALI, SII, LIPI, and mGPS were calculated before and after CRT, and their associations with PFS and OS after durvalumab consolidation were analyzed. Among these markers, mGPS before CRT (pre-CRT mGPS) had the highest C-index for both PFS (0.572) and OS (0.653) with statistical significance, suggesting a high predictive value for survival outcomes (Table 2). Furthermore, mGPS after CRT (post-CRT mGPS) had the highest C-index for PFS (0.549) and OS (0.615) among the variables investigated after CRT (Table 2). Therefore, mGPS was an essential marker in this study.

Unlike pre-CRT mGPS, post-CRT mGPS was not significantly associated with PFS or OS. The influence of persistent inflammation and body weight changes induced by CRT on post-CRT mGPS values was investigated because the mGPS consists of CRP and serum albumin levels. The numbers of patients with mGPS of 0, 1, and 2 at the initiation of CRT were 78 (61.9%), 26 (20.6%), and 22 (17.5%) patients, respectively. The mGPS was maintained at the same score after CRT in 76.9%, 11.5%, and 30.4% of patients with mGPS of 0, 1, and 2, respectively (Figure 1A). A negative correlation was observed (*r* = −0.301, *p* = 0.035) between CRP levels after CRT (post-CRT CRP levels) and body weight loss during CRT among patients with pre-CRT mGPS values of 1 or 2. However, no correlation existed between post-CRT CRP levels and body weight change during CRT in patients with pre-CRT mGPS values of 0 (Figure 1B). Thus, the extent of inflammation induced by CRT could affect post-CRT mGPS and body weight changes during CRT in patients with a pre-CRT mGPS of 1 or 2. This was not observed in patients whose pre-CRT mGPS values were 0. These results suggest the possibility of prognostic stratification based on the pre-CRT mGPS and persistent inflammation levels after CRT.

### 3.3. Systemic Inflammation-Based Prognostic Risk Classification in Patients Treated with Chemoradiotherapy Followed by Durvalumab Consolidation

We investigated the association between persistent inflammation based on post-CRT CRP levels and prognosis after durvalumab consolidation in each subgroup according to the pre-CRT mGPS. The inflammatory status after CRT was defined as low when CRP ≤1 mg/dL, and high when CRP >1 mg/dL, which is used as the CRP cutoff value in the mGPS [23,24]. Among the patients with mGPS of 0, no significant difference was observed in PFS and OS between 17 and 60 patients with high and low CRP levels, respectively, after CRT (hazard ratio [HR] for PFS: 1.07 [95% CI: 0.48–2.38], HR for OS: 1.51 [95% CI: 0.52–4.43]) (Figure 2A,B). In contrast, among the patients with mGPS of 1, 8 patients with high CRP levels after CRT had significantly poor PFS and relatively poor OS compared with 18 patients with low CRP levels (HR for PFS: 4.60 [95% CI: 1.44–14.7], HR for OS: 3.65 [95% CI: 0.79–16.84]) (Figure 2C,D). Among patients with mGPS of 2, 8 patients with high CRP levels after CRT had relatively shorter PFS and OS compared with 15 patients with low CRP levels without significance (HR for PFS: 1.12 [95% CI: 0.38–3.28], HR for OS 1.82 [95% CI: 0.51–6.52]) (Figure 2E,F).

These results suggest that the cohort with a pre-CRT mGPS of 1 consisted of patients with heterogeneous backgrounds, who could be divided into groups with good or poor outcomes based on the inflammation status after CRT. However, the cohorts with pre-CRT mGPS of 0 and 2 were considered relatively homogeneous and had good or poor outcomes irrespective of post-CRT inflammation status.

A prognostic risk classification reflecting systemic inflammation was created based on pre-CRT mGPS values and post-CRT CRP levels: patients with pre-CRT mGPS of 0 or mGPS of 1 with post-CRT CRP ≤1 mg/dL were classified as the “low-risk” group, and patients with pre-CRT mGPS of 2 or mGPS of 1 with post-CRT CRP >1 mg/dL were classified as the “high-risk” group. The high-risk group consisted of patients with more advanced stages than the low-risk group (*p* = 0.031) and experienced a significant weight loss during CRT compared with the low-risk group (low-risk vs. high-risk; −1.72% [IQR, −5.10, 0.76] vs. −4.86% [IQR, −8.24, −2.19], *p* = 0.037). The best overall response to CRT was not significantly different between the high- and low-risk groups (*p* = 0.859) (Table 3).

### 3.4. Relationship between Persistent Inflammation after Chemoradiotherapy and Survival Outcomes of Durvalumab Consolidation

The high-risk group had a significantly shorter median PFS of 7.2 months (95% CI: 4.5–19.3) compared with that of the low-risk group, which was 27.8 months (95% CI: 16.6-not reached [NR]) (HR: 2.47 [95% CI: 1.46–4.19], *p* < 0.001) (Figure 3A). Similarly, the median OS in the high-risk group was significantly shorter compared with that in the low-risk group (19.6 months [95% CI: 9.3–NR] vs. NR [95% CI: 31.7–NR], HR: 3.62 [95% CI: 1.79–7.33], *p* < 0.001) (Figure 3B).

In the multivariate analysis adjusted for age, sex, histology, and smoking history, the high-risk group with persistent inflammation had shorter PFS and OS (HR for PFS: 2.28 [95% CI: 1.27–4.07, *p* = 0.006], HR for OS: 3.48 [95% CI: 1.60–7.57], *p* = 0.002) (Figure 4A,B).

### 3.5. Impact of PD-L1 Expression on Progression-Free Survival during Durvalumab Consolidation

A subgroup analysis was performed on 91 (72.2%) patients with available PD-L1 expression status to evaluate the correlation between the impact of persistent inflammation and PD-L1 expression on tumor cells at baseline. In the subgroup with ≥50% PD-L1 expression on tumor cells, the high-risk group had significantly shorter PFS than the low-risk group (Figure 5A). Contrastingly, no significant difference existed between the high- and low-risk groups in the subgroup with <50% PD-L1 expression (Figure 5B).

### 3.6. Treatment-Related Adverse Events during Durvalumab Consolidation after Chemoradiotherapy

Treatment-related adverse events observed in ≥5% of the total population are presented in Table 4. Grade 3 or higher adverse events were more common in the high- than in the low-risk group (22.6% vs. 10.5%; *p* = 0.127). Additionally, toxicity-related treatment discontinuation was more frequent in the high- than in the low-risk group (35.7% vs. 20.5% of patients with terminated durvalumab administration).

## 4. Discussion

Preclinical studies have revealed that radiotherapy induces a potent antitumor immune response by inducing antigen presentation and the expression of class I major histocompatibility complexes, accompanied by the release of damage-associated molecular patterns [29]. The tumor microenvironment is easily altered by chemotherapy and radiotherapy. Tumors with increased levels of tumor-infiltrating lymphocytes (TILs) are associated with a better prognosis, whereas those with upregulated PD-L1 expression have a poorer prognosis [30]. Other chemoradiation-induced changes in the tumor microenvironment suggest that inflammatory cytokines are essential [31,32].

We evaluated immunological and nutritional markers to predict the PFS and OS of durvalumab consolidation after CRT and observed that pre- and post-CRT mGPS values were the essential predictors among the evaluated indices (Table 2). Furthermore, heterogeneity was observed among patients with a pre-CRT mGPS of 1 in terms of PFS and OS of durvalumab consolidation, differing among subgroups with post-CRT CRP levels of ≤1 mg/dL and >1 mg/dL (Figure 2C,D). Thus, a systemic inflammation-based prognostic risk classification was created by combining pre-CRT mGPS values and post-CRT CRP levels; the high-risk group had significantly shorter PFS and OS than the low-risk group (Figure 3A,B). This is the first report of an mGPS-based assessment of PFS and OS in durvalumab consolidation after CRT. While immunological and inflammatory markers, such as NLR [25] and CAR, [26] are potential predictive markers for durvalumab consolidation after CRT, this study revealed that mGPS, a combined marker of inflammation and nutrition, was strongly associated with survival outcomes than other markers.

The mechanism underlying the observed usefulness of combining pre-CRT mGPS and post-CRT CRP levels can be attributed to cancer cachexia, cancer-derived inflammation, and CRT-induced inflammation. The relationship between body weight change during CRT and post-CRT CRP levels was employed in determining the importance of elevated post-CRT CRP levels among patients with a pre-CRT mGPS of 1. mGPS, defined by serum CRP and albumin levels, is a simple and objective parameter for assessing cancer cachexia, focusing on nutrition and systemic inflammation. mGPS of 0, 1, and 2 correspond to non-cachexia, pre-cachexia, and cachexia, respectively [27]. CRT did not completely suppress tumor viability and tumor-derived inflammation in patients with persistent inflammation (i.e., elevated CRP levels) after CRT, because they relapsed earlier after durvalumab consolidation than those with improved inflammation (Figure 3A,B). However, evaluation of tumor viability at the time of CRT completion is challenging because of radiation-induced inflammation around the tumor and the time needed for tumor shrinkage after CRT. This retrospective study revealed that persistent inflammation after CRT was associated with shorter PFS and OS among patients with a pre-CRT mGPS of 1, but not patients with a pre-CRT mGPS of 0 or 2. This suggests that the inflammation induced by CRT would not lead to shorter PFS and OS in patients without inflammation prior to CRT (i.e., pre-CRT mGPS of 0) or cause further survival deterioration among patients with pre-existing cachexia (i.e., pre-CRT mGPS of 2). Therefore, tumor-derived inflammation, which partially reflects tumor viability, may be associated with poor therapeutic response to durvalumab consolidation after CRT and the development of cancer cachexia.

Cachexia is a poor prognostic factor for immunotherapy. The elevation of inflammatory cytokines, which is essential in cachexia development via the induction of skeletal muscle wasting and metabolic abnormalities, attenuates tumor immune responses, both directly and indirectly. IL-1β and tumor necrosis factor (TNF)-β reportedly contribute to forming the cold tumor immune microenvironment by suppressing TILs [33,34]. In contrast, elevated IL-6 levels suppress immune response activation by inhibiting glycogenesis in the liver and promoting glucocorticoid secretion [35]. Furthermore, inflammatory cytokines, including IL-6, IL-17, or TNF-α, upregulate PD-L1 expression [36,37]. Thus, inflammation might significantly impact the tumor response to PD-1/PD-L1 inhibition. Therefore, high PD-L1 expression on tumor cells—a good prognostic marker during anti-PD-1/PD-L1 therapy—does not invariably indicate a favorable antitumor response via PD-1/PD-L1 inhibition. This suggests that persistent inflammation that develops into cachexia or inappropriately increases PD-L1 expression may attenuate the immune response by PD-1/PD-L1 inhibition, even when PD-L1 is highly expressed on tumor cells. This could explain the shorter PFS of the high-risk group compared with that of the low-risk group in the subgroup with ≥50% PD-L1 expression on tumor cells (Figure 5A).

Another effect of systemic inflammation on survival outcomes is increased treatment-related adverse events. In patients with cachexia or pre-cachexia, elevated baseline levels of inflammatory cytokines and further inflammation due to treatment are associated with increased adverse events during immunotherapy [38]. In addition, patients with cachexia tend to have poor ECOG-PS and difficulty continuing chemotherapy, owing to adverse events [39]. Furthermore, platinum-based chemotherapy and chemoradiotherapy are associated with decreased plasma levels of ghrelin, an endogenous ligand for growth hormone secretagogue receptors. Ghrelin administration improves gastrointestinal symptoms [40,41]. Pharmacological intervention with anamorelin, a selective ghrelin receptor agonist, in addition to nutritional and exercise interventions, preserves and improves the nutritional status in patients with cachexia [42]. Thus, the early detection of cachexia or persistent inflammation in patients with locally advanced NSCLC treated with CRT reduces the risk of treatment discontinuation and maximizes the effect of durvalumab consolidation. This study suggests that patients with a pre-CRT mGPS of 1 and persistent systemic inflammation after CRT may experience early treatment discontinuation, owing to disease progression or treatment-related adverse events, which deteriorates survival outcomes.

This study has several limitations. First, the effect of unknown confounding factors could not be excluded because of the retrospective study design and limited sample size. For example, chronic obstructive pulmonary disease, smoking-related lung diseases, and the progression of atherosclerotic lesion would have led to the elevated CRP levels, affecting the results. Second, the data were based on medical records, rendering it challenging to accurately differentiate the source of persistent inflammation after CRT. Third, the cutoff values for various predictive markers were selected from previous reports. Therefore, using biomarkers and optimal cutoff values should be clarified in prospective studies with large sample sizes. The effect of non-pharmacological interventions, including nutritional management and exercise guidance, and pharmacological interventions for cachexia on the outcomes of durvalumab consolidation therapy should also be evaluated.

## 5. Conclusions

This study demonstrated that pre- and post-CRT mGPS values were the essential predictors of immunological and nutritional markers. Additionally, combining pre-CRT mGPS values and post-CRT CRP levels in patients with locally advanced NSCLC helped to predict the PFS and OS of durvalumab consolidation after CRT.

## Figures and Tables

**Figure 1 cancers-15-04358-f001:**
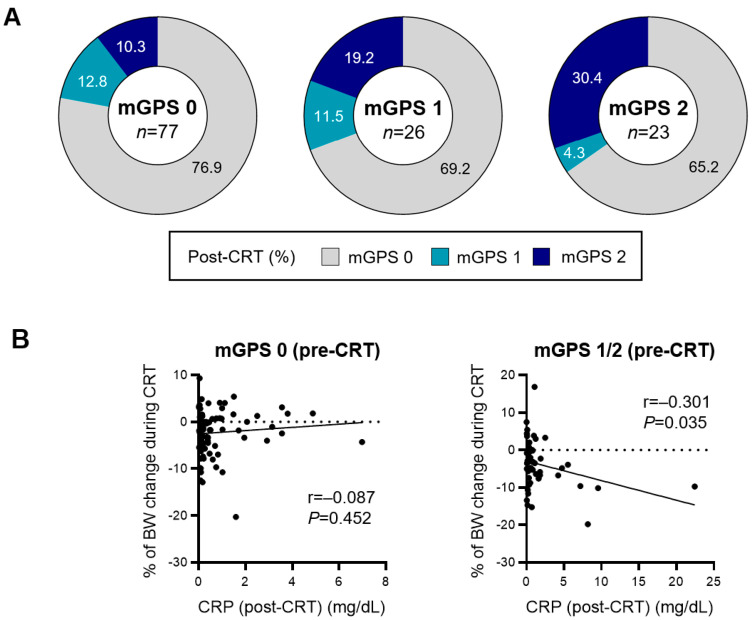
Change in modified Glasgow prognostic score before and after chemoradiotherapy. (**A**) The number of patients with each modified Glasgow prognostic score (mGPS) value before chemoradiotherapy (CRT) is indicated at the center of each circle. The percentage of patients with each mGPS after CRT is presented in the surrounding circled space with gray back (post-CRT mGPS of 0), blue back (post-CRT mGPS of 1), and navy back (post-CRT mGPS of 2). The mGPS values were maintained at the same score after CRT in 76.9%, 11.5%, and 30.4% of patients with mGPS of 0, 1, and 2, respectively. (**B**) Pearson correlation analysis for percentage of body weight change during CRT and C-reactive protein levels after CRT in the subgroup with mGPS 0 or 1/2 before CRT. There was a negative correlation (*r* = −0.301, *p* = 0.035) between CRP levels after CRT and body weight loss during CRT among patients with mGPS values of 1 or 2 before CRT.

**Figure 2 cancers-15-04358-f002:**
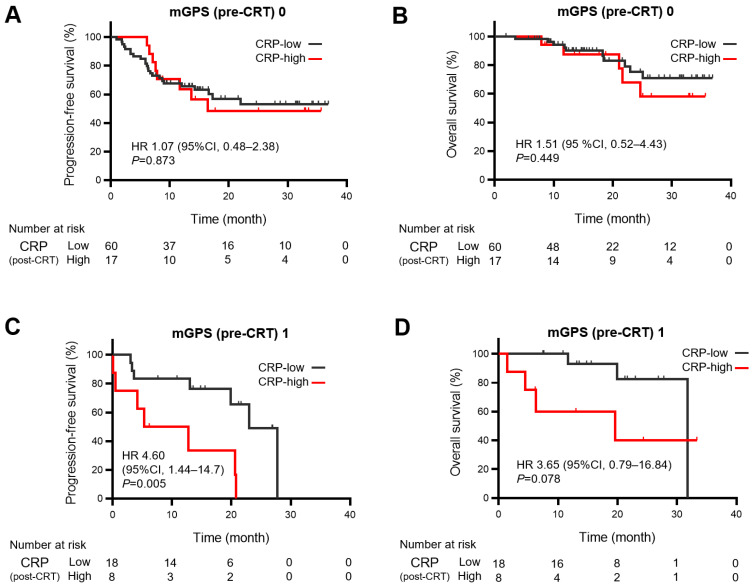
Kaplan–Meier estimates of progression-free survival and overall survival of durvalumab according to C-reactive protein levels (≤1 mg/dL vs. >1 mg/dL) after chemoradiotherapy (CRT) in subgroups based on modified Glasgow prognostic score before CRT. Kaplan–Meier estimates of (**A**) progression-free survival (PFS) and (**B**) overall survival (OS) in the subgroup with modified Glasgow prognostic score (mGPS) of 0, (**C**) PFS and (**D**) OS in the subgroup with mGPS of 1, and (**E**) PFS and (**F**) OS in the subgroup with mGPS of 2, based on C-reactive protein (CRP) levels of ≤1 mg/dL (low CRP level) and CRP >1 mg/dL (high CRP level). Among the patients with mGPS of 1, patients with high CRP levels after CRT had significantly poor PFS and relatively poor OS compared with patients with low CRP levels (HR for PFS: 4.60 [95% CI: 1.44–14.7], HR for OS: 3.65 [95% CI: 0.79–16.84]). HR, hazard ratio; CI, confidence interval.

**Figure 3 cancers-15-04358-f003:**
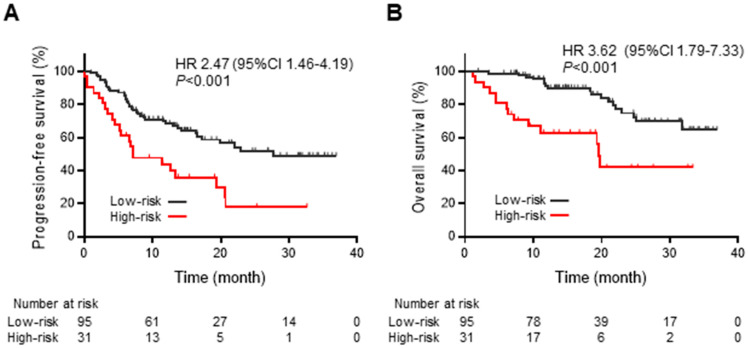
Kaplan–Meier estimates of progression-free survival and overall survival of durvalumab according to systemic inflammation-based prognostic risk classification. Kaplan–Meier estimates of (**A**) progression-free survival (PFS) and (**B**) overall survival (OS) according to the low- and high-risk groups based on the systemic inflammation-based prognostic risk classification. HR, hazard ratio; CI, confidence interval.

**Figure 4 cancers-15-04358-f004:**
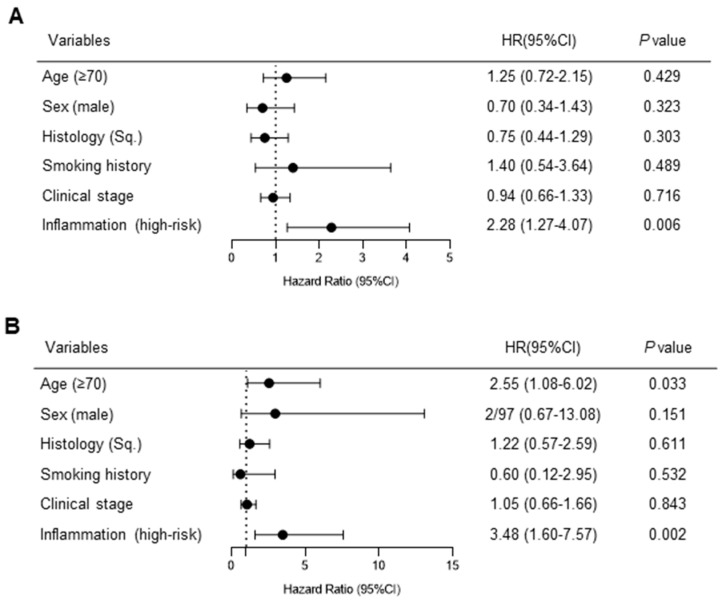
Multivariate analysis of (**A**) progression-free survival and (**B**) overall survival. In the multivariate analysis, persistent inflammation, defined as a pre-chemoradiotherapy (CRT)-modified Glasgow prognostic score (mGPS) of 2, or a pre-CRT mGPS of 1 and post-CRT C-reactive protein levels >1 mg/dL, was associated with shorter progression-free survival and overall survival. HR, hazard ratio; CI, confidence interval; Sq, squamous cell carcinoma.

**Figure 5 cancers-15-04358-f005:**
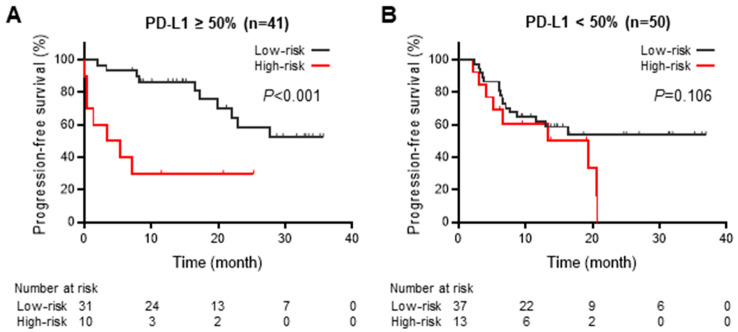
Treatment efficacy of durvalumab in subgroups with programmed cell death ligand 1 of ≥50% or <50% according to systemic inflammation-based prognostic risk classification. Kaplan–Meier estimates of progression-free survival of patients within high- and low-risk groups among the subgroup with (**A**) programmed cell death ligand 1 (PD-L1) ≥50% and (**B**) PD-L1 <50%, based on the systemic inflammation-based prognostic risk classification, using Fisher’s Exact Test.

**Table 1 cancers-15-04358-t001:** Characteristics of patients.

Characteristics		*n* = 126
Age—median (IQR), year		71.0 (64.3, 76.0)
Sex-Male—*n* (%)		98 (77.8)
Smoking history—*n* (%)	Never smoked	16 (12.7)
	Ex- or current smoker	110 (87.3)
ECOG-PS—*n* (%)	0 or 1	118 (93.7)
	≥2	8 (6.3)
Histology—*n* (%)	Squamous cell carcinoma	65 (51.6)
	Adenocarcinoma	57 (45.2)
	Others	4 (3.2)
Driver mutations—*n* (%)	Wild-type	19 (15.1)
	EGFR or ALK	6 (4.8)
	Others	2 (1.6)
	Unknown	99 (78.6)
PD-L1 expression—*n* (%)	≥50%	41 (32.5)
	<50%	50 (39.7)
	Unknown	35 (27.8)
Clinical stage—*n* (%)	≤IIb	11 (8.7)
	IIIA	48 (38.1)
	IIIB	56 (44.4)
	IIIC	11 (8.7)
Chemotherapy—*n* (%)	Carboplatin/paclitaxel	64 (50.8)
	Carboplatin monotherapy	22 (17.5)
	Cisplatin/S-1	16 (12.7)
	Cisplatin/docetaxel	14 (11.1)
	Others	10 (8.0)
Best overall response to CRT—*n* (%)	PR/CR	93 (73.8)
	SD	32 (25.4)
	NE	1 (0.8)
Interval of CRT and durvalumab -median (IQR), day		15.5 (13.0, 29.8)

IQR, interquartile range; ECOG-PS, Eastern Cooperative Oncology Group performance status; EGFR, epidermal growth factor receptor; ALK, anaplastic lymphoma kinase; PD-L1, programmed cell death ligand 1; CRT, chemoradiotherapy; PR, partial response; CR, complete response; SD, stable disease; S-1, tegafur/gimeracil/oteracil; NE, not evaluable.

**Table 2 cancers-15-04358-t002:** C-index of inflammation-based prognostic indicators in Cox proportional hazards model for progression-free survival and overall survival.

		PFS				OS			
	Pre/Post CRT	C-Index	SE	HR (95% CI)	*p*-Value	C-Index	SE	HR (95% CI)	*p*-Value
mGPS (0,1,2)	Pre	0.572	0.035	1.45 (1.07–1.97)	0.016	0.653	0.049	1.84 (1.22–2.78)	0.004
	Post	0.549	0.032	1.35 (0.99–1.83)	0.056	0.615	0.052	1.47 (0.98–2.20)	0.065
CAR (<0.32 vs. ≥0.32)	Pre	0.557	0.033	1.67 (1.0–2.77)	0.049	0.64	0.044	2.33 (1.16–4.68)	0.018
	Post	0.535	0.03	1.41 (0.81–2.45)	0.22	0.61	0.047	2.11 (1.03–4.33)	0.041
NLR (<5 vs. ≥5)	Pre	0.566	0.03	1.92 (1.04–3.55)	0.037	0.603	0.046	2.36 (1.09–5.11)	0.029
	Post	0.512	0.032	0.99 (0.58–1.69)	0.962	0.543	0.047	1.46 (0.71–3.01)	0.302
ALI (≥18 vs. <18)	Pre	0.553	0.032	1.56 (0.91–2.67)	0.103	0.604	0.047	2.43 (1.21–4.9)	0.013
	Post	0.489	0.034	1.02 (0.61–1.70)	0.931	0.526	0.048	1.24 (0.61–2.49)	0.556
PLR (<180 vs. ≥180)	Pre	0.507	0.034	1.04 (0.63–1.72)	0.879	0.532	0.048	1.12 (0.56–2.24)	0.756
	Post	0.54	0.028	0.64 (0.36–1.13)	0.122	0.54	0.043	0.69 (0.32–1.49)	0.34
SII (<750 vs. ≥750)	Pre	0.539	0.034	1.37 (0.82–2.27)	0.226	0.581	0.048	1.54 (0.76–3.12)	0.232
	Post	0.548	0.034	0.77 (0.46–1.27)	0.304	0.506	0.049	0.99 (0.49–1.98)	0.967
LIPI (0,1,2)	Pre	0.547	0.034	1.26 (0.86–1.84)	0.236	0.546	0.046	1.45 (0.85–2.47)	0.173
	Post	0.51	0.036	1.1 (0.73–1.67)	0.655	0.562	0.052	1.5 (0.87–2.59)	0.141

PFS, progression-free survival; OS, overall survival; CRT, chemoradiotherapy; SE, standard error; HR, hazard ratio; CI, confidence interval; mGPS, modified Glasgow prognostic score; CAR, C-reactive protein to albumin ratio; NLR, neutrophil-to-lymphocyte ratio; ALI, advanced lung cancer inflammation index; PLR, platelet-to-lymphocyte ratio; SII, systemic immune-inflammation index; LIPI, lung immune prognostic index.

**Table 3 cancers-15-04358-t003:** Characteristics of patients according to inflammation-associated prognostic risk classification.

		Low-Risk (*n* = 95)	High-Risk (*n* = 31)	*p*-Value
Age—median (IQR), y		70.0 [44.0, 89.0]	72.0 [36.0, 84.0]	0.351
Sex—*n* (%)	Male	75 (78.9)	23 (74.2)	0.622
ECOG-PS—*n* (%)	0 or 1	89 (93.7)	29 (93.5)	1
	≥2	6 (6.3)	2 (6.5)	
Smoking history—*n* (%)	Yes	83 (87.4)	27 (87.1)	1
	No	12 (12.6)	4 (12.9)	
Histology—*n* (%)	Non-Sq.	48 (50.5)	17 (54.8)	0.686
	Sq.	47 (49.5)	14 (45.2)	
PD-L1 expression—*n* (%)	≥50%	31 (32.6)	10 (32.3)	0.293
	1–49%	24 (25.3)	12 (38.7)	
	<1%	13 (13.7)	1 (3.2)	
	Unknown	27 (28.4)	8 (25.8)	
Clinical Stage	≤IIb	11 (11.6)	0 (0.0)	0.031
	IIIA	36 (37.9)	12 (38.7)	
	IIIB	43 (45.3)	13 (41.9)	
	IIIC	5 (5.3)	6 (19.4)	
% of change from pre-CRT BW—median (IQR)		−1.72 (−5.10, 0.76)	−4.86 (−8.24, −2.19)	0.037
Best overall response to CRT—*n* (%)	CR/PR	69 (72.6)	24 (77.4)	0.859
	SD	25 (26.3)	7 (22.6)	
	NE	1 (1.1)	0 (0)	

**Table 4 cancers-15-04358-t004:** Adverse events observed in 5% or more of the population and the relationship between high- and low-risk groups classified by inflammation-based prognostic risk classification.

	Total (*n* = 126)		Low-Risk (*n* = 95)		High-Risk (*n* = 31)	
Adverse Events—*n* (%)	Any Grade	Grade 3–5	Any Grade	Grade 3–5	Any Grade	Grade 3–5
Any AE	101 (80)	17 (13)	76 (80)	10 (10.5)	25 (80.6)	7 (22.6)
Pneumonitis	82 (65)	10 (8)	62 (65.2)	5 (5.3)	20 (64.5)	5 (16.1)
Hypothyroidism	19 (15)	5 (4)	16 (16.8)	4 (4.4)	3 (9.6)	1 (3.2)
Rash	11 (9)	0	9 (9.5)	0	2 (6.4)	0

AE, adverse event.

## Data Availability

The datasets generated and analyzed in the current study are available from the corresponding author upon reasonable request.

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
