# Peer review of "Predictive Value of Modified Glasgow Prognostic Score and Persistent Inflammation among Patients with Non-Small Cell Lung Cancer Treated with Durvalumab Consolidation after Chemoradiotherapy: A Multicenter Retrospective Study"

_cancers, 2023, doi:10.3390/cancers15174358_

Round 1

Reviewer 1 Report

The main question of this research was whether mGPS values before and after CRT are significant predictors of immunological and nutritional markers.   Early detection of cachexia or persistent inflammation reduces the need for treatment interruption and increases the consolidation effect of durvalumab.

The topic is original despite a retrospective study.   The study aims to identify predictors fo duravalumab consolidation after CRT, particularly focusing on systemic inflamation and cachexia to detect relaps without delay.

Compared with other published material,  this research's finds that early detection of cachexia or persistent inflammation reduces the need for treatment interruption and increases the consolidation effect of durvalumab.

Regarding the methodology, the authors could consider the inclusion affects the results of other chronic inflammations of the patients.

The conclusions are consistent with the evidence and arguments presented and they address the main question posed.

The references are appropriate.

No additional comments on the tables and figures.

Perhaps the researchers could obtain retrospective data from these patients regarding the presence of the most common chronic diseases such as COPD, generalized atherosclerosis, diabetes, heart failure, in which inflammatory markers may also be slightly elevated.

Author Response

Response:

I really appreciate your profound understanding of our manuscript as well as sincere suggestions.

As you pointed out, mGPS values both before and after chemoradiotherapy (CRT) were the essential predictors among investigated immunological and nutritional marker. In order to discriminate tumor-derived inflammation from CRT-induced inflammation, CRP levels after CRT were added to the pre-CRT mGPS.

I agree with you in that COPD or atherosclerosis could induce elevated CRP levels and affected the results of this study. Because it is difficult to fully elucidate the impact of these confounding factors.

Then, I added a sentence in the limitation “For example, chronic obstructive pulmonary disease, smoking-related lung diseases, and progression of atherosclerotic lesion would have led to the elevated CRP levels, affecting the results”.

Reviewer 2 Report

The Authors in this multicenter retrospective study aimed to identify the predictors of durvalumab consolidation after ChemoRadiotherapy (CRT) in lung patients cancer, especially focusing on systemic inflammation and cachexia, to detect recurrence without delay. The implications for Clinical Practice with a prognostic risk classification was created combining modified Glasgow Prognostic Score (mGPS) before CRT and C-reactive protein (CRP) level after CRT. When patients with pre-CRT mGPS of 0 or mGPS of 1 with post-CRT CRP ≤ 1 mg/dL were classified as the “low-risk” group, and patients with pre-CRT mGPS of 2 or mGPS of 1 with post-CRT CRP > 1 mg/dL were classified as the “high-risk” group, the high-risk group had a significantly shorter median progression-free survival  and overall survival  compared to those in the low-risk group.The study demonstrated that pre- and post-CRT mGPS values were the essential  predictors of immunological and nutritional markers. Additionally, combining pre-CRT mGPS values and post-CRT CRP levels in patients with locally advanced NSCLC helped to predict the PFS and OS of durvalumab consolidation after CRT. The early detection of cachexia or persistent inflammation reduces the risk of treatment discontinuation and maximizes the effect of durvalumab consolidation.

The study is well written and the statistical analysis is well conducted. The data are interesting and with clinical application. I believe the manuscript can be accepted for publication in the form in which it was submitted without further change.

No major modifications are required.

Author Response

Response:

I really appreciate your profound understanding of our manuscript. I am very pleased to hear your comments.

Reviewer 3 Report

Major issues:

1.      Material and Method,
2.1 Patients

The authors need to describe how the patients were enrolled in the current study even in a retrospective design. In other words, the inclusion and exclusion criteria should be disclosed. A patient selection flow chart (or diagram) is also suggested.

2.      Material and Method,
2.2. Immunological and nutritional markers
“Persistent inflammation” seems to be an important variable in this study, because the authors claim to investigate the predictive value of “persistent inflammation” for the treatment efficacy. In the circumstances, I’ll suggest the authors describe the measurement of “persistent inflammation” and explain its rationale and reference.

3.      Material and Method,
2.4 Statistics
Based on a retrospective nature, missing or unavailable data may be encountered. The authors may need to describe how to manage or handle the missing data once it is present.

4.      Conclusion
Line 390 to 392, “The early detection of ….. reduce the risk of ….. “
I don’t think the study results are supportive of this conclusion. Since the results only identify some predictive markers for the outcome. But it is an over-inference if you say the early detection of the markers could improve the outcome. This inference warrants further confirmation by prospective investigations.

Minor issues:

1.      Line 29, Line 43, and others in this manuscript.
I’ll suggest the authors use “compare with” but not “compare to.”

2.      Line 57.
Reference 4 is not supportive of the description. I found some better references, for example, PMID: 37454214 and PMID: 36602799.

3.      Line 111.
Here is a typo. When mGPS is 1, the CRP is >1 mg/dL or albumin <3.5 g/dL, not “≥” 3.5 g/dL.

4.      Line 241
In your result, patients with mGPS 0 or 2 are more “homogeneous” than those with 1.

Minor English editing may be required.

Author Response

I really appreciate your sincere suggestions and advice which is based on your precise understanding of our manuscript. I have carefully read through your comments, and prepared our responses in a point-by-point manner.

Major issues:

  1. Material and Method,
    1 Patients
    The authors need to describe how the patients were enrolled in the current study even in a retrospective design. In other words, the inclusion and exclusion criteria should be disclosed. A patient selection flow chart (or diagram) is also suggested.

Response:

I appreciate your suggestion. I added the inclusion and exclusion criteria in the Materials and Methods section (Lines 106-111). A patient selection flow diagram was attached in Supplementary Figure 1.

  1. Material and Method,
    2. Immunological and nutritional markers
    “Persistent inflammation” seems to be an important variable in this study, because the authors claim to investigate the predictive value of “persistent inflammation” for the treatment efficacy. In the circumstances, I’ll suggest the authors describe the measurement of “persistent inflammation” and explain its rationale and reference.

Response:

I appreciate your sincere suggestion. The importance of persistent inflammation after chemoradiotherapy was one of the main findings in this study. Therefore, this study was the first to investigate into this theme.

The systemic inflammation is well known to correlate with poorer survival outcomes in cancer patients, leading to cachexia. The immunological and nutritional markers investigated in this study also include inflammation markers, such as advanced lung inflammation index (ALI) and systemic immune inflammation index (SII).

I added the following sentence to emphasize on this point in the manuscript (Lines 121-122).

“CRP values were also investigated after CRT in order to evaluate the persistent inflammation after CRT.”

  1. Material and Method,
    4 Statistics
    Based on a retrospective nature, missing or unavailable data may be encountered. The authors may need to describe how to manage or handle the missing data once it is present.

Response:

I appreciate your important suggestion. Those patients with missing data were excluded from the analysis. I added this description in the Material and Method section (Line 150) as well as in the patient flow-diagram (Supplementary Figure 1).

  1. Conclusion
    Line 390 to 392, “The early detection of ….. reduce the risk of ….. “
    I don’t think the study results are supportive of this conclusion. Since the results only identify some predictive markers for the outcome. But it is an over-inference if you say the early detection of the markers could improve the outcome. This inference warrants further confirmation by prospective investigations.

Response:

I appreciate your sincere suggestion. I agree with you in that the last sentence in the Conclusion section needs more validation. Then, I deleted the relevant sentence.

Minor issues:

  1. Line 29, Line 43, and others in this manuscript.
    I’ll suggest the authors use “compare with” but not “compare to.”

Response:

I thank you for the suggestion. I have corrected accordingly.

  1. Line 57.
    Reference 4 is not supportive of the description. I found some better references, for example, PMID: 37454214 and PMID: 36602799.

Response:

I am sorry for the inappropriate reference. I changed the reference in accordance with your advice.

  1. Line 111.
    Here is a typo. When mGPS is 1, the CRP is >1 mg/dL or albumin <3.5 g/dL, not “≥” 3.5 g/dL.

Response:

I am sorry for the typographical error. I have corrected the relevant part accordingly.

  1. Line 241
    In your result, patients with mGPS 0 or 2 are more “homogeneous” than those with 1.

Response:

I am sorry for the typographical error. I have changed “heterogeneous” into “homogeneous”.